# From Anti-SARS-CoV-2 Immune Responses to COVID-19 via Molecular Mimicry

**DOI:** 10.3390/antib9030033

**Published:** 2020-07-16

**Authors:** Darja Kanduc

**Affiliations:** Department of Biosciences, Biotechnologies, and Biopharmaceutics, University of Bari, 70125 Bari, Italy; dkanduc@gmail.com

**Keywords:** peptide sharing, SARS-CoV-2 epitopes, molecular mimicry, cross-reactivity, autoimmunity

## Abstract

Aim: To define the autoimmune potential of Severe Acute Respiratory Syndrome Coronavirus 2 (SARS-CoV-2) infection. Methods: Experimentally validated epitopes cataloged at the Immune Epitope DataBase (IEDB) and present in SARS-CoV-2 were analyzed for peptide sharing with the human proteome. Results: Immunoreactive epitopes present in SARS-CoV-2 were mostly composed of peptide sequences present in human proteins that—when altered, mutated, deficient or, however, improperly functioning—may associate with a wide range of disorders, from respiratory distress to multiple organ failure. Conclusions: This study represents a starting point or hint for future scientific–clinical investigations and suggests a range of possible protein targets of autoimmunity in SARS-CoV-2 infection. From an experimental perspective, the results warrant the testing of patients’ sera for autoantibodies against these protein targets. Clinically, the results warrant a stringent surveillance on the future pathologic sequelae of the current SARS-CoV-2 pandemic.

## 1. Introduction

It is well known that coronavirus (re)infections have a pathologic immune potential. Indeed:Progressive immune-associated injury is a hallmark of SARS infection [1] and, since 2003 [2], the association with hyper-immune inflammation and systemic immunopathology in the SARS-CoV-infected host has been well illustrated [2,3,4,5];In 2008, it was reported [6] that prior immunization with SARS-CoV nucleocapsid protein N causes severe pneumonia in mice infected with SARS-CoV;In 2012, immunization with SARS coronavirus vaccines was found to lead to lung immunopathology on challenge with the SARS virus [7]; In 2016, Agrawal et al. [8] demonstrated that immunization with inactivated Middle East Respiratory Syndrome (MERS) coronavirus vaccine leads to lung immunopathology on challenge with live virus. Moreover, the Authors documented that immunopathology with SARS-CoV vaccines occurred for whole-virus vaccines, subunit vaccines, different inactivation methods, different preparation substrates, and with recombinant surface spike protein. Finally, the Authors also underscored that, even if studies with vector vaccines point to the nucleoprotein N as responsible for the immunopathological effects and indicate that the S protein might be free of risk, nonetheless, also recombinant spike protein induced the immunopathology [6,7,8,9].

However, the molecular and cellular basis for how SARS-CoVs might impact on the host immune system resulting in dysfunctional immune responses, severe morbidity, and mortality remain obscure [1,10].

Following the current SARS-CoV-2 pandemic, cross-reactivity was proposed as a mechanism that might explain the immunopathology associated with the coronavirus infection [11]. The rationale is that the sharing of peptide motifs between SARS-CoV-2 and human proteins might evoke immune responses capable of hitting not only the virus but also the human proteins with consequent autoimmune pathologic sequelae in the human host. As a matter of fact, the study [11] called attention to a vast and specific peptide sharing between SARS-CoV-2 spike glycoprotein and alveolar surfactant-related proteins, in this way addressing the issue of why SARS-CoV-2 so heavily attacks the respiratory system. 

In the wake of such results, in order to validate (or, as well, invalidate) the cross-reactivity hypothesis, investigation was expanded here by analyzing the peptide sharing between the human host and immunoreactive epitopes that are also present in SARS-CoV-2. Actually, the mere sharing of peptide sequences between pathogens and human proteins might be of little significance whether it remained sterile of cross-reactive autoimmune reactions. On the contrary, a peptide sharing between immunoreactive pathogen-derived epitopes and human proteins would assume the value of a proof of concept of cross-reactivity as a mechanism contributing to the pathogen-induced diseases. 

In this scientific framework, using the hexapeptide as an antigenic and immunogenic unit [12,13,14,15], immunoreactive epitopes that are present in SARS-CoV-2 were analyzed for matches with the human proteome. The results confirm a vast peptide commonality that involves human proteins implied in pulmonary insufficiency, neurological disorders, cardiac and vascular alterations, pregnancy dysfunctions, multiple cancers and anosmia, among others.

## 2. Materials and Methods

### 2.1. SARS-CoV-2 Epitopes

Analyses were conducted on an immunome composed of 233 linear epitopes that were experimentally validated, present in the proteins of SARS-CoV, and cataloged in the IEDB [16]. The experimentally validated 233 epitopes have been described in detail by Ahmed et al. [17], map identically in SARS-CoV-2 proteins, and are listed in Appendix A. 

The hexapeptide was used as a measurement unit to define minimal epitopic sequences. Indeed, 5–6 amino acids (aa) can form a sufficient minimal determinant for epitope–paratope interaction [12,13]. Likewise, T-cell epitopes contain a core of 5–6 aa that are involved in antigenicity and immunogenicity [13,14,15] and are flanked by NH_2_- and COOH– terminus aa that act as anchor motifs.

### 2.2. Analyses of the Peptide Sharing between SARS-CoV-2 Epitopes and the Human Proteome

The SARS-CoV-2 epitope sequences were dissected into hexapeptides which overlapped each other by five aa residues. The resulting 733 hexapeptides were analyzed for occurrence(s) within human proteins using the Pir Peptide Match program [18]. 

Human proteins involved in the hexapeptide sharing were analyzed for function/diseases using UniProtKB, PubMed, and OMIM resources.

### 2.3. Calculation of the Expected Value for Hexapeptide Sharing

The expected value for hexapeptide sharing between two proteins can be calculated by considering the number of all possible hexapeptides, *N*. Since in a hexapeptide each residue can be any of the 20 aa, the number of all possible hexapeptides *N* is given by *N* = 20^6^ = 64 × 10^6^. Then, the number of the expected occurrences is directly proportional to the number of hexapeptides in the two proteins and inversely proportional to *N*. Assuming that the number of hexapeptides in the two proteins is ≪*N* and neglecting the relative abundance of aa, we obtain a formula derived by approximation, where the expected number of hexapeptides is 1/*N* or 20^−6^.

## 3. Results

### 3.1. Numerical Description of the Peptide Sharing between SARS-CoV-2 Epitopes and the Human Proteome

Table 1 reports that 230 out of the 733 hexapeptides composing the analyzed 233 immuno- reactive epitopes occurred among 460 human proteins. Many of these shared sequences recurred more times. For instance, the zinc finger protein ZNF265—a splice factor that is important in renin mRNA processing and stability [19]—shares the hexapeptide SRSSSR with the viral epitope SQASSRSSSR (IEDB ID: 60380). The hexapeptide SRSSSR recurs three times in the zinc finger protein, exactly at aa positions: 211-216, 241-246, and 256-261. Then, including multiple occurrences, on the whole the hexapeptides shared with the human proteins amount to 505.

### 3.2. Distribution of the Shared Hexapeptides through the SARS-CoV-2 Epitopes

For reasons of synthesis, Table 2 reports the distribution of the shared hexapeptides relatively to a sample (*n* = 58) of SARS-CoV-2 epitopes. In the viral epitope sequences, the hexapeptides shared with human proteins are marked in capital letter format. The distribution of the shared hexapeptides throughout the entire set of 233 SARS-CoV-2 epitopes is described in Appendix A.

Table 2 documents that numerous immunoreactive SARS-CoV-2 epitopes are composed mostly or, in many instances, uniquely of peptide sequences shared with human proteins. As an example, among the many, the epitope IEDB ID 4936, which is present in the viral nucleocapsid protein and corresponds to the aa sequence ATEGALNTPK, results from the consecutive succession/overlapping of 6 mers present in human proteins too.

Taken together, Table 1, Table 2, and Appendix A highlight the unexpectedness of the peptide overlap between the human proteome and the immunoreactive viral epitopes derived from IEDB, described by Ahmed et al. [17] and analyzed here. From a mathematical point of view, if one considers that the probability of a hexapeptide to occur in 2 proteins is ~20^−6^ (or 1 out of 64,000,000 or 0.000000015625), then the number of hexapeptides (namely, 505, including multiple occurrences) shared between the SARS-CoV-2 immunome and human proteins is surprisingly high and can be explained only on the basis of evolutionary processes [20].

In the context of autoimmunity, Table 1, Table 2, and Appendix A concretize the effective possibility of cross-reactions between SARS-CoV-2 and the human proteome, given the fact that the immunological information unit in terms of both immunogenicity and antigenicity is contained in a space formed by 5–6 aa residues [12,13,14,15].

### 3.3. Distribution of the Hexapeptide Sharing through the Human Proteins and the Potential Diseasome

As quantified in Table 1, hexapeptides from immunoreactive viral epitopes occur across 460 human proteins. The 460 human proteins are listed and synthetically described in Appendix A. The 460 human proteins are involved in metabolic, developmental, and regulatory cellular functions and—when mutated, modified, deficient or, however, improperly functioning—may lead to altered functions and more or less severe pathologies in the human organism. Obvious reasons of space prevent a protein-by-protein analysis of all of them and, here, only a few proteins (given by the UniProt name in italic and shared peptides in parentheses) and the related pathologies that could arise in case of cross reactivity are dealt with as follows.


**Pulmonary disorders**


Molecular mimicry between SARS-CoV-2 spike glycoprotein and alveolar surfactants-related proteins have already been described [11]. Additional proteins that–if hit–can alter lung functions are:
*Mothers against decapentaplegic homolog 9 protein* (QASSRS), alterations of which may lead to pulmonary hypertension with proliferating endothelial cells in pulmonary arterioles, right ventricular failure, and death [21];*Phosphatidylinositol 4,5-bisphosphate 3-kinase catalytic subunit gamma isoform* (IKDLPK) that is deficient in pulmonary vascular endothelial cells of patients with acute respiratory distress syndrome [22,23];*Adenylate cyclase type 9* (KQLSSN) that is expressed in multiple cells of the lung with expression highest in airway smooth muscle [24];*Acetylcholinesterase* (AVLQSG) where imbalances in the neurotransmitter acetylcholine relate to neurological conditions, such as Alzheimer’s disease, Parkinson’s disease, and myastenia gravis; irreversible inhibition of acetylcholinesterase may lead to muscular paralysis, convulsions, bronchial constriction, and death by asphyxiation [25].


**Cancer of the lung and other organs:**
*Endoribonuclease Dicer* (ASSRSS) which is linked to pleuropulmonary blastoma [26];*Protein naked cuticle homolog 1* (LLPSLA), downregulation of which is linked to poor prognosis in non-small-cell lung cancer and breast invasive ductal carcinoma [27,28];*Downregulated in multiple cancers 1* (AAEIRA) is not expressed in multiple human cancers [29];*Ubiquitin carboxyl-terminal hydrolase BAP1* (LLSVLL) relates to tumor predisposition syndrome; linked to mesothelioma [30,31,32];In addition, the tumor suppressor proteins *pinin* (SSRSSS) [33], *tripartite motif-containing protein 35* (SFKEEL) [34], *unconventional myosin-IXb* (LLPSLA) [35], and *cyclin-D-binding myb-like transcription factor 1* (AALQIP) [36].



**Cardiac disorders:**
*Low-density lipoprotein receptor-related protein 8* (LALLLL), alterations of which can lead to myocardial infarction [37];*Dol-P-Glc:Glc(2)Man(9)GlcNAc(2)-PP-Dol alpha-1,2-glucosyltransferase* (TLTLAV) is implicated in susceptibility to the long QT syndrome [38];*Presenilin-2* (TLACFV) relates to dilated cardiomyopathy and heart failure [39];*Nesprin-1* (LLSAGI) is involved in dilated or hypertrophic cardiomyopathy [40,41]:*Nuclear receptor coactivator 6* (PSLATV) can cause dilated cardiomyopathy [42];*Latent-transforming growth factor beta-binding protein 3* (LALLLL) can associate with skin thickening, cardiac valvular thickening, tracheal stenosis, and respiratory insufficiency [43].



**Vascular disorders:**
*Ephrin-B3* (LALLLL) is involved in blood pressure control and vascular smooth muscle cell contractility [44];*Endoglin* (LVLSVN) is required for normal structure and integrity of adult vasculature [45];*Elastin* (FGAGAA) is a major structural protein of tissues, such as aorta and nuchal ligament, and relates to cutis laxa disease, may also, although rarely, lead to pulmonary artery stenosis, aortic aneurysm, bronchiectasis, and emphysema [46,47];*Filamin-A* (PYRVVV), alterations of *filamin-A* can cause disorders related to the vascular system and to a large phenotypic spectrum of disorders such as deafness, urogenital defects, malformations, intestinal obstruction, constipation, recurrent vomiting, and diarrhea [48];*Glomulin* (RIMASL) is crucial in vascular morphogenesis—especially in cutaneous veins [49].



**Autoinflammatory syndrome**
**:**
*Tumor necrosis factor receptor superfamily member 1A* (GTTVLL) may associate with fever, abdominal pain, localized tender skin lesions, myalgia, and reactive amyloidosis as the main complication [50,51].



**Coagulopathies:**
*Thrombomodulin* (AGAALQ), a well-known natural anti-coagulant that acts as a linker of coagulation and fibrinolysis, is involved in disseminated intravascular coagulation, is a predictor of risk of arterial thrombosis and is essential for the maintenance of pregnancy [52,53,54].In addition, cross-reactive peptide sharing involving additional proteins can further affect the level of thrombomodulin. Indeed, cross-reactions with *r**etinoic acid receptor RXR-alpha* (LGFSTG) and *prostaglandin G/H synthase 2* (TVLLKE) might completely eliminate thrombomodulin from blood circulation because (1) *retinoic acid receptor RXR-alpha* promotes the thrombomodulin gene transcription [55] and (2) *prostaglandin G/H synthase 2* stimulates the expression of functionally active thrombomodulin in human smooth muscle cells [56].Adding up to such pro-thrombotic scenario, it is also worth of mention the potential cross-reactivity with *tyrosine-protein kinase JAK2* (LLDDFV) that is involved in myelofibrosis, myeloid leukocytosis, and thrombocytosis with excessive platelet production resulting in increased numbers of circulating platelets, hemorrhages, and thrombotic episodes [57,58,59].



**Neurological disorders:**
*Neuronal PAS domain-containing protein 2* (ASSRSS), which is highly polymorphic in autism spectrum disorder patients [60,61];*Circadian locomotor output cycles protein kaput* (ASSRSS) relates to bipolar disorder [62];*Adenosine receptor A1* (VLPPLL), where sleep is significantly attenuated by the loss of adenosine A1 receptor expression [63];*BDNF/NT-3 growth factors receptor* (SANLAA), alterations may cause temporal lobe epilepsy, memory impairment, anorexia nervosa, bulimia, Alzheimer’s disease [64,65,66];*Calcium/calmodulin-dependent protein kinase kinase 2* (PSLATV) is linked to disturbances of higher cognitive functions, such as working memory and executive function. as well as schizophrenia [67];*Endoplasmic reticulum mannosyl-oligosaccharide 1,2-alpha-mannosidase* (LAFLLF) can cause mental retardation [68];*Glutaminase kidney isoform, mitochondrial* (LQELGK) can associate with epileptic encephalopathy, infantile cataract, skin abnormalities leukocytoclasia at the surface of the dermis, focal vacuolar alterations, hyperkeratosis, parakeratosis, glutamate excess, and impaired intellectual development, global developmental delay, progressive ataxia, and elevated glutamine [69,70,71];*Mitochondrial glutamate carrier 1* (RLQSLQ) relates to neonatal myoclonic epilepsy [72].


Moreover, and remarkably, the *voltage-gated calcium channel gamma subunits* 2, 3, 4, 6, and 8 contain epitopic hexapeptides (Table 3). Voltage-gated calcium channels control cellular calcium entry in response to membrane potential changes [73,74], and gamma subunits 2, 3, 4, and 8 regulate α-Amino-3-hydroxy- 5-Methyl-4-isoxazolePropionic Acid Receptor (or AMPA receptor or AMPAR) and are collectively known as Transmembrane AMPAR Regulatory Proteins (TARPs) [75]. AMPARs are involved in the fast synaptic transmission in the central nervous system and are altered in many psychological and neurological disorders such as schizophrenia, depression and Parkinson’s disease [76].


**Anosmia:**


As a final note, this Results Section closes with one disorder that is one of the first symptoms to appear in COVID-19, namely, anosmia. The role of anosmia as a first symptom is clearly justified by the fact that seven olfactory receptors (i.e., proteins related to the smell) [77] share hexapeptides with the viral epitopes (Table 4).

## 4. Discussion

This study shows that hexapeptides from immunoreactive epitopes present in SARS-CoV-2 are widespread among a high number of human proteins. Such a peptide sharing implies the possibility of cross-reactions and, consequently, as discussed in the Results (Section 3), of a vast phenotypic constellation of diseases, from pneumonia and neurological disorders to cardio-vascular alterations and coagulopathies. Hence, this study appears to offer scientific hints to explain the clinical fact that SARS-CoV-2 infection is capable of triggering so many and so different pathologies in so many and so different organs of the human host [78].

On the whole, the data indicate that—besides possible virus-induced multi-organ direct cytopathic effects and other possible pathogenic mechanisms—self-reactive antibodies may be at the root of the pathologic scenario that accompanies SARS-CoV-2 infection. In fact, previous research focusing on SARS-CoV, which similarly to SARS-CoV-2 causes a respiratory failure syndrome, showed that anti-spike protein IgGs can cause lung damage by directly affecting inflammatory mechanisms, promoting the release of pro-inflammatory cytokines, such as IL-8, as well as macrophagic tissue infiltration [79]. A similar effect of SARS-CoV-2 on inflammatory response has already been proposed [80]. Then it appears reasonable to hypothesize that autoantibody-mediated macrophagic and complement activation and autoantibody-dependent cell-mediated cytotoxicity mechanisms might be some of the mechanisms behind the well-documented multi-organ damage in COVID-19, thus representing one of many possible links between adaptive and innate immunity in the pathogenesis of the disease.

Moreover, as reviewed by Vabret et al. [81], it has been shown that high antibody response can associate with more severe clinical cases. This had also been seen in the previous SARS-CoV-1 epidemic, where neutralizing antibody titers were found to be significantly higher in deceased patients compared to patients who had recovered [82] and might lead to suspect antibody-dependent enhancement phenomena.

Of note, the present data also indicate that most possibly the damage caused by SARS-CoV-2 might not end with the end of the pandemic. Indeed, the peptide sharing between SARS-CoV-2 epitopes and specific tumor suppressor proteins, and, in general, many tumor-related proteins [26,27,28,29,30,31,32,33,34,35,36], theoretically predicts, in the absence of current clinical data, that a morbidity/mortality increase in various cancers might follow the current SARS-CoV-2 pandemic.

As a conclusive note, it is mandatory also to observe that this study largely undervalues the potential risk of cross-reactivity between SARS-CoV-2 immunome and human proteins. Indeed, analyses were conducted on linear epitopes and used the hexapeptide as an immune unit probe. Then, if one considered conformational epitopes and used the pentapeptide as a minimal immune determinant [13,83], the number of viral versus human commonalities would increase exponentially. Given this premise, the present results call researchers and clinicians for a common research effort to study the autoimmune pathogenicity connected to the anti-SARS-CoV-2 immune response and suggests, once more, that using entire antigens in anti-SARS-CoV-2 vaccine formulations might lead to autoimmune manifestations and adverse events [84]. Actually, the present data further support the fundamental concept that only “non-self” peptides can lead to safe and efficacious immunotherapies [85,86,87].

## Figures and Tables

**Table 1 antibodies-09-00033-t001:** Hexapeptide sharing between 233 epitopes present in SARS-CoV-2 and human proteins.

Hexapeptides composing the 233 epitopes	733
Hexapeptides shared with the human proteome	230
Hexapeptides shared with the human proteome (including multiple occurrences)	505
Human proteins involved in the sharing	460

**Table 2 antibodies-09-00033-t002:** SARS-CoV-2 epitopes with sequences shared with human proteins marked in capital letters.

A ^1^	B ^2^	C ^3^	A ^1^	B ^2^	C ^3^
956	N	AEGSRGGSQA	37,515	N	lLLLDRLNql
999	S	aeiRASANLA	37,544	S	lLLQYGSfc
1220	S	aevqidrli	37,583	orf1b	Llmpiltlt
1221	S	aevqidrlit	37,724	S	LLQYGSfct
1349	ORF7b	AFLLFLVLI	37,766	orf1a	LLSAGIFGA
1350	ORF7b	AFLLFLVLIMLIIFw	38,043	orf1b	Lmierfvsl
1946	orf1a	AIILASFSA	38,353	S	lntLVKQLSSNFGAi
2027	orf1b	aimtrclav	38,831	S	LQDVVNQNAQALNTL
2431	N	ALALLLLDr	38,855	S	Lqipfamqm
2682	orf1b	ALLADKfpv	38,874	orf1b	LQLGFSTGv
2801	S	alntlvkql	38,881	N	LQLPQGttl
2802	N	ALNTPKdhi	38,990	S	lqslqtyvtQQLIRA
2855	orf1a	aLRANSAvk	39,003	S	lqtyvtQQLIRAAEI
2998	orf1a	alweiqqvv	39,576	N	Lsprwyfyy
3589	S	aphgvvflhv	40,459	orf1b	LVLSVNpyv
3810	N	APSASAFFgm	40,677	M	Lwllwpvtl
3939	S	aQALNTLvk	40,685	M	lWPVTLAcf
3956	N	AQFAPSASA	41,962	ORF7b	mLIIFWFSL
3982	S	aqkfnGLTVLPPLLT	42,093	orf1b	Mlwckdghv
4307	N	asaffgmsr	42,128	orf1b	Mmisagfsl
4321	S	ASANLAATk	42,260	orf1a	Mpaswvmri
4936	N	ATEGALNTPK	42,648	N	msriGMEVTPSGTWl
5149	M	aTSRTLSYY	42,873	S	mTSCCSCLk
5150	M	aTSRTLSYYK	42,972	orf1b	mvMCGGSLyv
5209	orf1b	ATVVIGtsk	43,024	M	mwSFNPETni
5447	orf1a	AVLQSGFRK	44,501	N	nkhidayktFPPTEP
5908	S	ayrfngiGVTQNVly	44,814	S	nlnESLIDL
6184	ORF7a	celyhyqecv	44,913	M	nlVIGFLFL
6668	S	cmTSCCSCLk	60,380	N	sQASSRSSSR

**^1^** A: Epitopes listed as IEDB ID number and detailed at IEDB [16]. **^2^** B: SARS-CoV-2 proteins [18]. **^3^** C: Hexapeptides shared with human proteins are given in capital letters.

**Table 3 antibodies-09-00033-t003:** Hexapeptide sharing between SARS-CoV-2 epitopes and TARPs.

Shared Hexapeptide(s)	TARP
LSAGIF, SRSSSR	gamma-2 subunit
LSAGIF	gamma-3 subunit
SRSSSR	gamma-4 subunit
LGAGCF	gamma-6 subunit
SRSSSR	gamma-8 subunit

**Table 4 antibodies-09-00033-t004:** Hexapeptide sharing between SARS-CoV-2 epitopes and olfactory receptors.

Shared Hexapeptide	Olfactory Receptor
AIILAS	2B11
AIILAS	2W1
SVLLSM	51G1
SVLLSM	51G2
RMLLEK	51J1
IFWFSL	52N5
IIFWFSL	7D4

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
