# Peer review of "From Anti-SARS-CoV-2 Immune Responses to COVID-19 via Molecular Mimicry"

_2073-4468, 2020, doi:10.3390/antib9030033_

Round 1
Reviewer 1 Report
The work in this manuscript is only a starting point or hint for future investigation either experimentally or clinically. The data in current manuscript is not sufficient.
Author Response
1st Referee
The work in this manuscript is only a starting point or hint for future investigation either experimentally or clinically. The data in current manuscript is not sufficient.
Answer: Thanks for recognizing the usefulness of the presented data as a starting point for future investigations. Indeed the paper aims at presenting potentially cross-reactive epitopes to the scientific community in order to facilitate the testing of the hypothesis that autoimmunity contributes to Covid-19 pathogenesis.
In this second revision, this point has been specified under Abstract (see, please, lines 17 and 18)

Reviewer 2 Report
my comments have been addressed
the question regarding number of references will need to be settled with the journal Editor
Author Response
my comments have been addressed
the question regarding number of references will need to be settled with the journal Editor
Answer: Thanks for the insightful comments.
The long list of references has been shortened and 12 references have been eliminated (see please, lines 485-521) in the present 2nd Revision

Reviewer 3 Report
Ref. antibodies-850522
This is the revised version of the manuscript “From anti-SARS-CoV-2 immune responses to COVID 19 via molecular mimicry”, by Darja Kanduc, previously submitted under # antibodies-831-351. The author reports a computational analysis of peptide sharing between the SARS-CoV-2 and human proteomes, and the implications for the pathogenesis of Covid-19. In the new version, the author made textual changes that addressed some of the concerns raised by the other reviewers.
My criticism to the first version of the manuscript could be summarised into the following points:
- Lack of experimental validation;
- Bias in the selection of proteins for discussion;
- Failure to discuss alternative (and more likely) mechanisms for Covid-19 immunopathogenesis;
- Dual use research of concern.
To justify the lack of follow-up validation of the predicted cross-reactivity between the SARS-CoV-2-specific response and human proteins [point (1)], the author argues the listing of potentially cross-reactive epitopes per se might be useful for others to test the hypothesis that autoimmunity contributes to Covid-19 pathogenesis. This is a valid point, counting that the limitations of an in silico analysis are clearly stressed in the text, and strong statements (Page 2/Lines 60-63) or overarching conclusions (Page 7/Lines 262-264) are not made. In the rebuttal letter, to explain how specific antibody binding would lead to the malfunction of the target protein, the author makes a comparison with the impact that single nucleotide variations can have on protein structure and function. That is not a valid parallel. Key to the interpretation of the predictions made in the manuscript is to discuss how auto-reactive antibodies and T cells might contribute to the disease caused by SARS-CoV-2. In the absence of experimental validation that auto-antibodies/T cells are induced and that they contribute to the disease, this discussion should be based on the known facts about the mechanisms of infection and disease by human coronaviruses, and on hypotheses that are biologically plausible.
Regarding the point (2), the author argues in the rebuttal letter the analysis that retrieved the 460 human proteins that shared identity with immunoreactive 6-mer epitopes of SARS-CoV-2 is unbiased. I agree with this, but my point concerned to the ~50 proteins that were highlighted in the text. It still isn’t clear which criteria were used to select these 50 proteins out of the 460 flagged up by the computational analysis.
On to the point (3). Severe cases of Covid-19 have been linked to hyper-inflammation due to cytokine overexpression. In this work, the author hypothesises that autoimmunity might play a key role in Covid-19, but still in the revised version fails to discuss and contextualise their predictions with the clinical observations and the more widely accepted model for the disease. How does the response elicited by the shared epitopes contribute to the onset and (or) severity of Covid-19 symptoms?
My last major criticism concerned the potential “dual use” [point (4)] of the work. Here, the author deemed my objection contradictory and I would like to clarify it. Whereas I am of the opinion that the present work does not further our knowledge about autoimmunity triggered by viral infections, I stand for my concern that, by associating SARS-CoV-2 to cancer and foetal alterations, this work might provide knowledge (in the sense of information published in an academic peer-reviewed journal) to feed the sensationalist media and conspiracy theorists. Though the author has removed the statement directly associating SARS-CoV-2 to cancer and foetal alterations from the abstract, there should be clear statements in the remaining of the text about the lack of clinical data linking SARS-CoV-2 to cancer and foetal alteration.
Author Response
3rdReferee
This is the revised version of the manuscript “From anti-SARS-CoV-2 immune responses to COVID 19 via molecular mimicry”, by Darja Kanduc, previously submitted under # antibodies-831-351. The author reports a computational analysis of peptide sharing between the SARS-CoV-2 and human proteomes, and the implications for the pathogenesis of Covid-19. In the new version, the author made textual changes that addressed some of the concerns raised by the other reviewers.
My criticism to the first version of the manuscript could be summarised into the following points:
- Lack of experimental validation;
- Bias in the selection of proteins for discussion;
- Failure to discuss alternative (and more likely) mechanisms for Covid-19 immunopathogenesis;
- Dual use research of concern.
To justify the lack of follow-up validation of the predicted cross-reactivity between the SARS-CoV-2-specific response and human proteins [point (1)], the author argues the listing of potentially cross-reactive epitopes per se might be useful for others to test the hypothesis that autoimmunity contributes to Covid-19 pathogenesis. This is a valid point, counting that the limitations of an in silico analysis are clearly stressed in the text, and strong statements (Page 2/Lines 60-63) or overarching conclusions (Page 7/Lines 262-264) are not made. In the rebuttal letter, to explain how specific antibody binding would lead to the malfunction of the target protein, the author makes a comparison with the impact that single nucleotide variations can have on protein structure and function. That is not a valid parallel. Key to the interpretation of the predictions made in the manuscript is to discuss how auto-reactive antibodies and T cells might contribute to the disease caused by SARS-CoV-2. In the absence of experimental validation that auto-antibodies/T cells are induced and that they contribute to the disease, this discussion should be based on the known facts about the mechanisms of infection and disease by human coronaviruses, and on hypotheses that are biologically plausible.
Answer: Thanks for the productive and stimulating discussion and for recognizing some relevant points of my replies. Strong statements and overarching conclusions have been removed. As the Reviewer suggests, key known facts about the mechanisms of infection and disease by human coronaviruses, and relevant hypotheses that are biologically plausible and originate from the present work are discussed below, please see reply to point (3).
Regarding the point (2), the author argues in the rebuttal letter the analysis that retrieved the 460 human proteins that shared identity with immunoreactive 6-mer epitopes of SARS-CoV-2 is unbiased. I agree with this, but my point concerned to the ~50 proteins that were highlighted in the text. It still isn’t clear which criteria were used to select these 50 proteins out of the 460 flagged up by the computational analysis.
Answer: It was considered the possible importance of the potentially hittable protein. For example the now deleted proteins that, when altered, might relate to fetal/infantile disorders seemed of interest in light of the clinical literature that suggests “further spatial-temporal studies to determine the true potential for SARS-CoV-2 transplacental migration.” [Mahyuddin AP, et al. Mechanisms and evidence of vertical transmission of infections in pregnancy including SARS-CoV-2 Prenat Diagn 2020 doi:10.1002/pd.5765]. This in agreement also with The Royal Colleges of Obstetricians and Gynaecologists, Midwives, Paediatrics and Child Health that, at an astonishing speed, reconfigured guidance services towards virtual care, guide safeguarding, and emphasise social distancing for pregnant women and vulnerable children 2020. https://www.gov.uk/government/publications/covid-19-guidanceon-social-distancing-and-for-vulnerable-people].
It was also considered the numerical criterion. That is, anosmia is illustrated with seven olfactory receptor proteins, of which all have been listed (see Table 4) since referenced by 1 single reference [see ref.77]. Instead, proteins linked to already treated disorders such as pulmonary disorders [11], were not discussed.
On to the point (3). Severe cases of Covid-19 have been linked to hyper-inflammation due to cytokine overexpression. In this work, the author hypothesises that autoimmunity might play a key role in Covid-19, but still in the revised version fails to discuss and contextualise their predictions with the clinical observations and the more widely accepted model for the disease. How does the response elicited by the shared epitopes contribute to the onset and (or) severity of Covid-19 symptoms?
Answer: Thanks for the question. Previous research focussing on SARS-CoV, that similarly to SARS-CoV-2 causes a respiratory failure syndrome, showed how anti-spike protein IgGs can cause lung damage by directly affecting inflammatory mechanisms, promoting the release of pro-inflammatory cytokines such as IL-8 as well as macrophagic tissue infiltration (Liu et al., Anti–spike IgG causes severe acute lung injury by skewing macrophage responses during acute SARS-CoV infection, JCI Insight 4:2019). A similar effect of SARS-CoV-2 on inflammatory response has already been proposed (Fu et al. Understanding SARS-CoV-2-mediated inflammatory re-sponses: from mechanisms to potential therapeutic tools, Virol. Sin. 2020). It appears then reasonable to hypothesize that autoantibody-mediated macrophagic and complement activation and autoantibody-dependent cell-mediated cytotoxicity mechanisms might be some of the mechanisms behind the well-documented multi-organ damage in COVID-19, thus representing one of many possible links between adaptive and innate immunity in the pathogenesis of the disease.
My last major criticism concerned the potential “dual use” [point (4)] of the work. Here, the author deemed my objection contradictory and I would like to clarify it. Whereas I am of the opinion that the present work does not further our knowledge about autoimmunity triggered by viral infections, I stand for my concern that, by associating SARS-CoV-2 to cancer and foetal alterations, this work might provide knowledge (in the sense of information published in an academic peer-reviewed journal) to feed the sensationalist media and conspiracy theorists. Though the author has removed the statement directly associating SARS-CoV-2 to cancer and foetal alterations from the abstract, there should be clear statements in the remaining of the text about the lack of clinical data linking SARS-CoV-2 to cancer and foetal alteration.
Answer: Thanks for clarifying. I definitely agree that the media are prone to misinterpret the nature of scientific information not being able to tell theory, hypothesis and degree of evidence apart. That said, scientists should not censor their science just to avoid being misunderstood. Nevertheless, given the Reviewer’s concern in this particularly critic moment for society, the data relative to the infantile/foetal alterations and the relative 12 references have been removed (see lines 204-217 and lines 266-268). As far as cancer is regarded, the phrase “in the absence of current clinical data” has been added in the last but one paragraph of Conclusions (line 265).

Round 2
Reviewer 1 Report
The authors attempted to simply revise the text to address reviewers's comments, but the study was definitely not substantially improved or validated experimentally. The information they presented was apparently biased (host proteins and associated diseases they discussed) and may potentially mislead authors or non-scientist audience.
Author Response
1st Referee criticism:
The authors attempted to simply revise the text to address reviewers's comments, but the study was definitely not substantially improved or validated experimentally. The information they presented was apparently biased (host proteins and associated diseases they discussed) and may potentially mislead authors or non-scientist audience.
ANSWER:
The study is an in silico identification of SARS-CoV-2 epitopes, that is standard practice in immunology (please, see: Grifoni A, Sidney J, Zhang Y, Scheuermann RH, Peters B, Sette A. A Sequence Homology and Bioinformatic Approach Can Predict Candidate Targets for Immune Responses to SARS-CoV-2. Cell Host Microbe. 2020;27(4):671-680.e2. doi:10.1016/j.chom.2020.03.002; and Shivarov V, Petrov PK, Pashov AD. Potential SARS-CoV-2 Preimmune IgM Epitopes. Front Immunol. 2020;11:932. doi:10.3389/fimmu.2020.00932 for recent examples), and that, in this specific case, focuses on known immunoreactive epitopes showing molecular mimicry with human proteins and potentially involved in autoimmunity.
It is here important to highlight that the epitopes that I described have all been previously experimentally validated as immunoreactive and catalogued in the Immune Epitope DataBase (IEDB), which is a freely available resource funded by the National Institute of Allergy and Infectious Diseases( NIAID) in Bethesda.. This step is even missing from many standard in silico studies that have been published and adds relevance to the data I present. Of course, I do agree with the Reviewer that these epitopes need to be investigated in COVID-19 and this is exactly the reason why these data might provide useful information to authors and scientists. I honestly do not see how authors and scientists can be misled by my proposal of testing these epitopes and addressing the issue of autoimmunity in COVID-19.
3rd Referee criticism:
In the rebuttal to the point (3) of my last review, the author discusses how the response elicited by the shared epitopes could contribute to the pathogenesis of Covid-19. I recommend the inclusion of this information in the discussion.
The information was included in the discussion, see please lines 245-254.
Recently, a team from the Mount Sinai Hospital reviewed the state-of-the-art of the immunology of Covid-19 (https://doi.org/10.1016/j.immuni.2020.05.002). The author should discuss how the hypothesised autoimmunity elicited by shared epitopes fits in the big picture portrayed in this review.
Thanks for the suggestion. Autoimmunity in the context of a broader picture of immunology of Covid-19 is now reported in this 3rd Revision in lines 255-259
Another point that should be discussed is how SARS-CoV-2 compare to other human pathogenic viruses in regards to epitope sharing between viral and human proteomes. Is the frequency of shared epitopes higher in SARS-CoV-2?
This is an important observation. Currently, one main problem hindering such a comparison is the size of the available epitope immunomes since a few pathogens have been intensively and extensively studied (see for instance HIV-1 with 706 validated IEDB epitopes) whilst others such as Ebola virus have only 184 validated IEDB epitopes. A comparison based on this unbalanced representation of experimentally validated IEDB epitopes would therefore be, from a practical point of view, methodologically incorrect, at least given the currently available resources.
Page 2, line 88: 206 = 6.4 x 107.
Page 2, line 88 (nowline 86) was corrected as : N = 206 = 64 × 106
That is, the total number of possible hexapeptides is 64 millions.
In Table 2, the footnote states that "Hexapeptides shared with human proteins are given in capital letters", but some of the shared epitopes have more than 6 aa residues. Also in this table, it's confusing that epitopes that aren't shared with human proteins start with a capital letter, which is used to denote the shared peptides.
Thanks for the corrections.

Reviewer 3 Report
This is the revised version of the manuscript # antibodies-831-351, reporting a computational analysis of peptide sharing between the SARS-CoV-2 and human proteomes, and the implications for the pathogenesis of Covid-19. In the most recent version, the author again made textual changes to address the concerns raised by me.
By toning down the reach of its conclusions, I believe the manuscript now could be published.
Minor points:
- In the rebuttal to the point (3) of my last review, the author discusses how the response elicited by the shared epitopes could contribute to the pathogenesis of Covid-19. I recommend the inclusion of this information in the discussion. Recently, a team from the Mount Sinai Hospital reviewed the state-of-the-art of the immunology of Covid-19 (https://doi.org/10.1016/j.immuni.2020.05.002). The author should discuss how the hypothesised autoimmunity elicited by shared epitopes fits in the big picture portrayed in this review.
- Another point that should be discussed is how SARS-CoV-2 compare to other human pathogenic viruses in regards to epitope sharing between viral and human proteomes. Is the frequency of shared epitopes higher in SARS-CoV-2?
- Page 2, line 88: 206 = 6.4 x 107.
- In Table 2, the footnote states that "Hexapeptides shared with human proteins are given in capital letters", but some of the shared epitopes have more than 6 aa residues. Also in this table, it's confusing that epitopes that aren't shared with human proteins start with a capital letter, which is used to denote the shared peptides.
Author Response
3rd Referee criticisms:
In the rebuttal to the point (3) of my last review, the author discusses how the response elicited by the shared epitopes could contribute to the pathogenesis of Covid-19. I recommend the inclusion of this information in the discussion.
The information was included in the discussion, see please lines 245-254.
Recently, a team from the Mount Sinai Hospital reviewed the state-of-the-art of the immunology of Covid-19 (https://doi.org/10.1016/j.immuni.2020.05.002). The author should discuss how the hypothesised autoimmunity elicited by shared epitopes fits in the big picture portrayed in this review.
Thanks for the suggestion. Autoimmunity in the context of a broader picture of immunology of Covid-19 is now reported in this 3rd Revision in lines 255-259
Another point that should be discussed is how SARS-CoV-2 compare to other human pathogenic viruses in regards to epitope sharing between viral and human proteomes. Is the frequency of shared epitopes higher in SARS-CoV-2?
This is an important observation. Currently, one main problem hindering such a comparison is the size of the available epitope immunomes since a few pathogens have been intensively and extensively studied (see for instance HIV-1 with 706 validated IEDB epitopes) whilst others such as Ebola virus have only 184 validated IEDB epitopes. A comparison based on this unbalanced representation of experimentally validated IEDB epitopes would therefore be, from a practical point of view, methodologically incorrect, at least given the currently available resources.
Page 2, line 88: 206 = 6.4 x 107.
Page 2, line 88 (now line 86) was corrected as : N = 206 = 64 × 106
That is, the total number of possible hexapeptides is 64 millions.
In Table 2, the footnote states that "Hexapeptides shared with human proteins are given in capital letters", but some of the shared epitopes have more than 6 aa residues. Also in this table, it's confusing that epitopes that aren't shared with human proteins start with a capital letter, which is used to denote the shared peptides.
Thanks for the corrections.

This manuscript is a resubmission of an earlier submission. The following is a list of the peer review reports and author responses from that submission.
Round 1
Reviewer 1 Report
In this manuscript, Darja Kanduc reports a computational analysis of peptide sharing between the SARS-CoV-2 and human proteomes and discusses the possible implications of it to the pathogenesis of Covid-19. Peptide sharing between microbes and hosts has been long known, including multiple reports from the author’s group. However, the physio-pathological relevance of peptide sharing remains largely underexplored and the current manuscript does little to further our knowledge beyond generating a list of 460 human proteins that share linear hexapeptides with SARS-CoV-2 proteins.
Following, the author highlighted some proteins associated to diseases when mutated or malfunctioning. Nonetheless, it wasn’t clear which criteria were used for the selection of the highlighted proteins, and whether the pathological conditions derived from the occurrence of a specific autoimmune response. An unbiased gene ontology analysis of the 460 proteins on http://geneontology.org/ didn’t retrieve any significant enrichment of particular biological processes or molecular functions either, undermining the interpretation that autoimmunity due to peptide sharing between virus and host might explain some of the Covid-19 pathogenesis. Without experimental corroboration of such interpretation, this work won’t contribute to our fundamental understanding of SARS-CoV-2 and Covid-19.
The author also falls short of exploring alternative mechanisms for the immunopathogenesis of Covid-19 other than autoimmunity due to peptide sharing. Moreover, the author makes strong and dangerous claims in linking SARS-CoV-2 to cancer fetal/infantile alterations based solely on an in silico analysis. That might configure “dual use research of concern” by providing information that could be misapplied, potentially jeopardising the global efforts to control the Covid-19 pandemics.
I thus don’t endorse the publication of these findings as an Original Research article without experimental validation.
Reviewer 2 Report
good and timely research. conclusions are well stated. few comments that i believe need to be addressed prior to the acceptance are listed here
- analysis was done using linear epitopes present on the virus proteins. it should be clarified earlier on that this will cover the T-cell epitopes only. Author proposed that B-cell epitopes are also included in the analysis although this is typically assumed that linear epitopes are representing T-cell repertoire. The author should be upfront with this important detail of the analysis.
- analysis of T-cell epitopes is frequently over-predicting. many more epitopes may be proposed while in reality these are more theoretical in nature and don't materialize in practice. should be clarified so there is no confusion
- has the author attempted to assess antigenicity of the hexapeptides identified as shared with human proteome in the analysis?
- clarify what is meant by "hexapeptides shared with human proteome (including multiple occurrences)". what is meant by multiple occurrences? multiple proteins?
- sentence on lines 84-85 is confusing. What is capital format? this may be obvious to the author, not necessarily to the reader.
- consider explaining terminology used and data presented in Table 2 better. this can be obvious the author but not so for a broader readership that you likely want to attract
- authors should consider expanding on analysis of other virus sequence to compare and benchmark. what are the general probability that viral genome can have such cross to human proteome? there is some discussion about general statistics of occurrence of 6-peptides which should be explained better
- citation index is quite long. justified?
Reviewer 3 Report
The manuscript by the author does not have a proper structure, a coherent language, real data, or an adequate interpretation of experimental data.
Reviewer 4 Report
In this study, the authors attempted to define the autoimmune potential of SARS-CoV-2 infection by comparing epitopes present in SARS-CoV2 with human proteome. Overall it's a computational study and the conclusion is not validated experimentally and should not be over-interpreted.
- To verify methodology used in this study, it is necessary to include other viruses or pathogens which are known not to cause autoimmune pathology in human. Compare their potential epitopes with human proteome.
- A large body of evidence show the immunopathology of coronavirus infection is associated with cytokine storm in acute phase. The author should explain how cross-reactivity of particular epitope leads to cytokine storm.
- If the cross-reactivity hypothesis is true, chronic organ pathology should be observed in patients after recovery.